# The Association between Diabetic Retinopathy and Macular Degeneration: A Nationwide Population-Based Study

**DOI:** 10.3390/biomedicines12040727

**Published:** 2024-03-25

**Authors:** Hsin-Ting Lin, Cai-Mei Zheng, Cheng-Hung Tsai, Ching-Long Chen, Yu-Ching Chou, Jing-Quan Zheng, Yuh-Feng Lin, Chia-Wei Lin, Yong-Chen Chen, Chien-An Sun, Jiann-Torng Chen

**Affiliations:** 1Department of Ophthalmology, Tri-Service General Hospital, National Defense Medical Center, Taipei 114, Taiwan; t72010@yahoo.com.tw (H.-T.L.); doc30881@mail.ndmctsgh.edu.tw (C.-L.C.); 2Graduate Institute of Medical Sciences, National Defense Medical Center, Taipei 114, Taiwan; linyf@s.tmu.edu.tw; 3Division of Nephrology, Department of Internal Medicine, School of Medicine, College of Medicine, Taipei Medical University, Taipei 110, Taiwan; 11044@s.tmu.edu.tw; 4TMU Research Center of Urology and Kidney, Taipei Medical University, Taipei 110, Taiwan; 5Division of Nephrology, Department of Internal Medicine, Shuang Ho Hospital, Taipei Medical University, Taipei 235, Taiwan; 6Graduate Institute of Clinical Medicine, College of Medicine, Taipei Medical University, Taipei 110, Taiwan; 16044@s.tmu.edu.tw; 7School of Public Health, National Defense Medical Center, Taipei 114, Taiwan; martintsai0920@yahoo.com.tw (C.-H.T.); trishow@mail.ndmctsgh.edu.tw (Y.-C.C.); 8Division of Pulmonary Medicine, Department of Internal Medicine, Shuang Ho Hospital, Taipei Medical University, Taipei 110, Taiwan; 9Department of Urology, Shuang Ho Hospital, Taipei Medical University, New Taipei City 110, Taiwan; sky02578@yahoo.com.tw; 10Department of Urology, School of Medicine, College of Medicine, Taipei Medical University, Taipei 110, Taiwan; 11Department of Urology, Tri-Service General Hospital, National Defense Medical Center, Taipei 114, Taiwan; 12Department of Medicine, College of Medicine, Fu Jen Catholic University, New Taipei City 242, Taiwan; 137159@mail.fju.edu.tw; 13Department of Public Health, College of Medicine, Fu Jen Catholic University, New Taipei City 242, Taiwan

**Keywords:** diabetes mellitus (DM), diabetic retinopathy (DR), non-diabetic retinopathy (non-DR), age-related macular degeneration (AMD), Taiwan National Health Insurance Database (NHIRD)

## Abstract

Objective: Age-related macular degeneration (AMD), particularly its exudative form, is a primary cause of vision impairment in older adults. As diabetes becomes increasingly prevalent in aging, it is crucial to explore the potential relationship between diabetic retinopathy (DR) and AMD. This study aimed to assess the risk of developing overall, non-exudative, and exudative AMD in individuals with DR compared to those without retinopathy (non-DR) based on a nationwide population study in Taiwan. Methods: A retrospective cohort study was conducted using the Taiwan National Health Insurance Database (NHIRD) (2000–2013). A total of 3413 patients were placed in the study group (DR) and 13,652 in the control group (non-DR) for analysis. Kaplan–Meier analysis and the Cox proportional hazards model were used to calculate the hazard ratios (HRs) and adjusted hazard ratios (aHRs) for the development of AMD, adjusting for confounding factors, such as age, sex, and comorbid conditions. Results: Kaplan–Meier survival analysis indicated a significantly higher cumulative incidence of AMD in the DR group compared to the non-DR group (log-rank test, *p* < 0.001). Adjusted analyses revealed that individuals with DR faced a greater risk of overall AMD, with an aHR of 3.50 (95% CI = 3.10–3.95). For senile (unspecified) AMD, the aHR was 3.45 (95% CI = 3.04–3.92); for non-exudative senile AMD, it was 2.92 (95% CI = 2.08–4.09); and for exudative AMD, the aHR was 3.92 (95% CI = 2.51–6.14). Conclusion: DR is a significant risk factor for both overall, senile, exudative, and non-exudative AMD, even after adjusting for demographic and comorbid conditions. DR patients tend to have a higher prevalence of vascular comorbidities; however, our findings indicate that the ocular pathologies inherent to DR might have a more significant impact on the progression to AMD. Early detection and appropriate treatment of AMD is critically important among DR patients.

## 1. Introduction

Diabetic retinopathy (DR) and age-related macular degeneration (AMD) are important retinal degenerative disorders that represent a growing concern among aging societies. These conditions exhibit overlapping pathological processes, such as retinal edema and progressive inflammation within the central macula. The hyperglycemic environment of diabetes mellitus activates inflammatory cells and increases the production of reactive oxygen species (ROS) and pro-inflammatory cytokines (e.g., IL-6, TNF-α, macrophage migration inhibitory factor (MIF), etc.) [1,2]. In addition, diabetes is characterized by the elevated release of substances, such as angiotensin II, prostaglandins (PGs), and vascular endothelial growth factor (VEGF), all of which are implicated in causing retinal vascular alterations [3]. These inflammatory and vascular changes are common underlying mechanisms that are pivotal in the progression of DR and AMD [4,5,6]. 

The balance between pro-inflammatory and anti-inflammatory pathways is crucial in managing the progression of both DR and AMD in diabetic patients. An imbalance between these pathways, marked by a deficiency in anti-inflammatory bioactive lipids and an accumulation of inflammatory cytokines, exacerbates the progression of these conditions [7,8,9]. Retinal pigment epithelial (RPE) cells, which protect against AMD and DR, counteract the production of ROS, inflammatory cytokines, and adhesion molecules through the secretion of pigment epithelium-derived factor (PEDF) [10,11]. However, in the context of diabetes, hyperglycemia disrupts the function of these RPE cells, impeding their PEDF production, which, in turn, can initiate retinal edema, DR, and AMD [12,13].

In Taiwan, an aging society, there is an increasing prevalence of both DR and AMD among elderly diabetic patients. While DR and AMD share common pathophysiological pathways, the epidemiological relationship between DR and AMD is still a subject of debate. Some studies have identified a link between DM and AMD [14,15], while others have not observed a significant correlation [16,17]. DM itself related positively with early AMD among elderly Korean patients [18], whereas a similar relation was noted between DR and neovascular AMD in a multicenter, population-based European study [19]. Herein, we used our population-based database to explore the incidence and risks for different types of AMD among patients with and without DR. 

## 2. Methods

### 2.1. Data Source

Taiwan’s National Health Insurance (NHI) was launched by the Taiwan Department of Health in 1995, providing comprehensive medical care coverage. A total of 23,832,551 Taiwanese residents, accounting for approximately 99% of the population in Taiwan, have joined the program. The Longitudinal Health Insurance Database (LHID), which is part of the NHI Research Database (NHIRD), contains data for one million randomly sampled patients from the Registry for Beneficiaries of the NHIRD. The database exhibits no statistically significant differences in terms of age, sex, or healthcare costs when compared to all NHI enrollees. Diagnoses in the database were assigned by qualified clinical physicians based on laboratory, imaging, and pathological data and following the International Classification of Diseases, Ninth Revision, Clinical Modification (ICD-9-CM). Personal identification numbers were encrypted to safeguard patient privacy before the electronic files were released for study.

### 2.2. Study Population

We conducted a population-based retrospective cohort study based on the LHID 2000. Following the ICD-9-CM format, AMD has several types, such as senile (unspecified), non-exudative, exudative, familial, juvenile, congenital, cystic AMD, etc. (Table 6), and we chose the most often cited codes by Ophthalmologists, which are senile (unspecified) (362.50) AMD, non-exudative (362.51) AMD, and exudative (362.52) AMD, to represent all types of AMD for reasonable convenience of further analysis. This study design will result in inconsistency of codes cited by Ophthalmologists and will be listed as a study limitation. We selected patients (aged ≥55 years old) who were newly diagnosed with diabetic retinopathy (DR) using ICD-9-CM codes 362.01 and 362.02 between 2000 and 2006 as the study cohort. To only include new cases, individuals who had previously received any DR and diabetes mellitus (DM) diagnoses, using ICD-9-CM code 250, in the medical claim data before 2000 were excluded. Additionally, DR patients with a history of age-related macular degeneration (AMD) using ICD-9-CM codes 362.50, 362.51, and 362.52 before the index date were excluded (*n* = 211). For each patient diagnosed with DR, we randomly selected four patients without diabetic retinopathy (non-DR) as the control group from the same database who were matched for age and sex. The non-DR group also excluded individuals with a history of AMD before the index date. Overall, our final sample for analysis included 3413 subjects with DR (the study group) and 13,652 subjects without DR (the control group). The study flowchart is shown in Figure 1. Both the study and control groups were followed to detect the occurrence of AMD. All subjects were observed until a diagnosis of AMD, death, or 31 December 2013, whichever occurred first. Our study was approved by the Institutional Review Board of Fu Jen Catholic University (FJU-IRB NO: C104014).

Histories of comorbidities present in both the study group and control group, such as hypertension (ICD-9 codes 401, 402, 403, 404, and 405), hyperlipidemia (ICD-9-CM codes 272.0, 272.1, 272.2, and 272.4), coronary artery disease (CAD; ICD-9-CM codes 410, 411, 412, 413, and 414), stroke (ICD-9-CM codes 430, 431, 432, 433, 434, 435, 436, 437, and 438), chronic obstructive pulmonary disease (COPD; ICD-9-CM codes 490, 491, 492, 493, 494, 495, and 496), and liver cirrhosis/chronic hepatitis (ICD-9-CM code 571), were included in this study to control for their potential confounding effects. 

### 2.3. Statistical Analysis

Chi-square tests were used to analyze categorical variables, including demographics and comorbidities, while continuous variables were assessed with two-sample *t*-tests. Kaplan–Meier curves were used to depict cumulative AMD incidence and Cox regression was used to estimate AMD risk, adjusted for age, sex, and comorbidities and stratified by age and sex to determine DR and AMD association. Data were analyzed via SAS 9.4 and SPSS 22.0, with *p* < 0.05 set as the significance level.

## 3. Results

The baseline demographic information and comorbidity conditions of patients in the diabetic retinopathy (DR) group (*N* = 3413) and the non-diabetic retinopathy (non-DR) group (*N* = 13,652) are presented in Table 1. The mean age in the DR group (67.1 years) was significantly higher than that in the non-DR group (66.5 years) (*p* < 0.001); however, there were no significant differences in age and gender between groups. DR patients had significantly higher rates of comorbidities, including hypertension, hyperlipidemia, coronary artery disease (CAD), stroke, chronic obstructive pulmonary disease (COPD), and liver cirrhosis/chronic hepatitis, compared to the non-DR group (all *p* < 0.001) (Table 1). Table 2 shows the occurrence and hazard ratios of all types of AMD for both the DR and non-DR groups. There was a higher incidence rate of AMD (16.77 per 10,000 person years (PYs)) in the DR group compared to the incidence rate of 5.36 per 10,000 PYs in the non-DR group. After adjusting for age, sex, and comorbid conditions, individuals with DR still had a significantly higher risk (aHR 3.50, 95% CI = 3.10–3.95) of overall AMD compared to those without DR (see Table 2). The incidence of AMD increased with age in both groups, but the association was stronger in the DR group across all age categories, with the highest risk observed in patients over 85 years old. The incidence rates of AMD were higher for both genders in the DR group, with males showing a slightly higher risk than females. Markedly, the DR group had a significantly higher HR and aHR than the non-DR group, regardless of its comorbid conditions.

Table 3 shows the comparative risks of senile (unspecified) AMD based on DR status. The non-DR group (*n* = 13,652) had an incidence rate of 4.78 per 10,000 PYs compared to the DR group (*n* = 3413) with an incidence rate of 14.65 per 10,000 PYs. The aHR for senile AMD in the DR group was 3.45 (95% CI: 3.04–3.92) compared to the non-DR group. Aging and the male gender were correlated with significantly higher risks than other demographic factors. A significant increase in AMD was noted in the DR group regardless of the presence or absence of comorbid conditions, such as hypertension, CAD hyperlipidemia, stroke, COPD, or cirrhosis/chronic hepatitis. These findings substantiate DR as a significant predictor of senile AMD, independent of demographic factors and comorbid conditions. Table 4 examines the incidence and hazard ratios of non-exudative senile AMD in DR compared to non-DR groups. A higher incidence rate of non-exudative AMD was observed in the DR group (1.89 per 10,000 person years) compared to the non-DR group (0.69 per 10,000 person years). The incidence rates increased with age in both the DR and non-DR groups. In the DR group, the incidence rate was markedly higher with aging, particularly in the 65–75 and ≥85 age groups. Like in overall AMD, male patients with DR had a higher incidence rate of non-exudative AMD compared to female patients with DR. Patients in DR had significantly higher incidence rates of AMD than their non-DR counterparts, regardless of their comorbid conditions (Table 4). 

A notably higher incidence rate of exudative AMD was observed in the DR group (1.37 per 10,000 person years) compared to the non-DR group (0.31 per 10,000 person years). The incidence rates of exudative AMD increased with age in both groups. However, the DR group exhibited significantly higher rates than non-DR over all age groups, especially in the 55–65 and 65–75 age groups. Males in the DR group had a higher incidence rate of exudative AMD compared to females in the same group. After adjusting for confounding factors, the aHRs of exudative AMD remained significantly higher in the DR group, and the risks were influenced by the presence of cardiovascular comorbid conditions, such as hypertension, hyperlipidemia, and CAD (Table 5). Table 6 presents types of age-related macular degeneration derived from ICD-9-CM Volume 2 Index entries containing backreferences to 362.50. Figure 2, Figure 3, Figure 4 and Figure 5 illustrate the higher cumulative risk of all types of macular degeneration, senile (unspecified) AMD, non-exudative AMD, and exudative AMD in the DR group compared to that of the non-DR group.

## 4. Discussion

This population-based cohort study explored the risks of age-related macular degeneration (AMD) in senile (unspecified), exudative, and non-exudative forms in elderly patients with diabetic retinopathy (DR). The initial assessment of baseline characteristics, including age and gender, showed similar distributions between the DR and non-DR groups, as detailed in Table 1. This eliminates the potential impact of these variables on our findings. However, a notably higher prevalence of comorbidities, such as hypertension, hyperlipidemia, and coronary artery disease (CAD), was observed in patients with DR. These findings are in line with previous research underscoring the multifaceted nature of diabetes, with its extensive impact on various vascular diseases [20]. These vascular comorbidities are known to contribute to damage in retinal small vessel endothelial cells, thereby influencing the development and severity of DR, and might also have an impact on the development of AMD [21,22].

Our study revealed that DR patients have a significantly increased incidence of AMD, even after accounting for other variables. This higher incidence might be attributed to the chronic inflammation associated with DR, which leads to an increased expression of vascular endothelial growth factor (VEGF), a key factor in the development of AMD [23]. Consistent with previous studies, our research indicates that age is a significant risk factor for AMD, particularly in individuals older than 85, possibly due to hormonal variations, genetic predispositions, or other unknown factors [24,25,26]. Accordingly, we observed a higher occurrence of senile (unspecified) AMD in the DR group compared to the non-DR group, as shown in Table 3. The risk of AMD increases with age and is higher in men, regardless of DR status. Comorbid conditions, such as hypertension, hyperlipidemia, and CAD, further increase the risks of AMD. It appears that a complex interaction between DR and other systemic diseases influences the risk of AMD [27]. Overall, the data emphasize the need for regular eye examinations for AMD in individuals with DR. 

Furthermore, individuals with DR were found to have a significantly higher likelihood of developing both non-exudative and exudative AMD compared to those without DR, as presented in Table 4 and Table 5. Age and the male gender were identified as risk factors in both groups. The pathogenesis of exudative AMD often involves significant vascular components, which could be exacerbated by the presence of DR [28]. Although comorbidities like hypertension, hyperlipidemia, and CAD typically heighten the risks for AMD, their relative impact is less pronounced within the DR patient group. This suggests that DR-related pathophysiological changes might play a more dominant role in AMD development than individual systemic conditions. Chronic inflammation, oxidative stress, and impaired vascular function in DR are responsible for structural changes in the retinal pigment epithelium (RPE), which, in turn, increases the risk of AMD [29,30]. Pathways such as VEGF expression [31], impaired lipoprotein metabolism [32], and mitochondrial dysfunction [33] are common in the development of both DR and AMD. Our study highlights the importance of regular AMD monitoring and assessment among DR patients, irrespective of comorbid conditions. 

The strength of our study lies in its comprehensive evaluation of demographic and comorbidity factors. This study included a large population sample size and examined various demographic variables, such as age and sex, as well as several comorbidities, including hypertension, hyperlipidemia, CAD, stroke, COPD, and liver cirrhosis and chronic hepatitis. Adjusting for these variables in the analysis not only enhances the robustness of the findings but also strengthens the evidence of the positive association between DR and AMD risk. 

However, as our study is a retrospective data analysis, it is subject to inherent biases and limitations, including potential selection bias and the inability to establish causality. On the other hand, most Ophthalmologists cited senile (unspecified) AMD, which includes non-exudative AMD and exudative AMD, which results in large differences in patient numbers among different types of AMD. This inconsistency of citation by Ophthalmologists might result in a large difference in number among different types of AMD. Additionally, the study findings have been limited by the specified population and are not generalizable to other populations. 

In conclusion, our study revealed a multifaceted relationship between DR and AMD. DR patients are more likely to have vascular conditions, which can contribute to the onset of all types of AMD. Here, we firstly provided the relation between DR and AMD, regardless of comorbid conditions, andsuggested that the inherent pathologies related to DR might have a greater impact on the progression to AMD than comorbid conditions. This is in contrast to the non-DR cohort, where a clearer correlation between comorbidities and the risk of AMD is observed. However, our study diverges in its emphasis on the heightened risk of non-exudative AMD in DR patients, a less explored aspect in previous studies. This discrepancy underscores the complexity of its association and the imperative for continued exploration in this field. Nevertheless, this study highlights the critical need for regular ocular monitoring in individuals with DR, with a particular focus on detecting AMD. Healthcare providers should maintain a heightened awareness of the increased risk of AMD in the DR population and emphasize the need for proactive and preventive eye care measures.

## Figures and Tables

**Figure 1 biomedicines-12-00727-f001:**
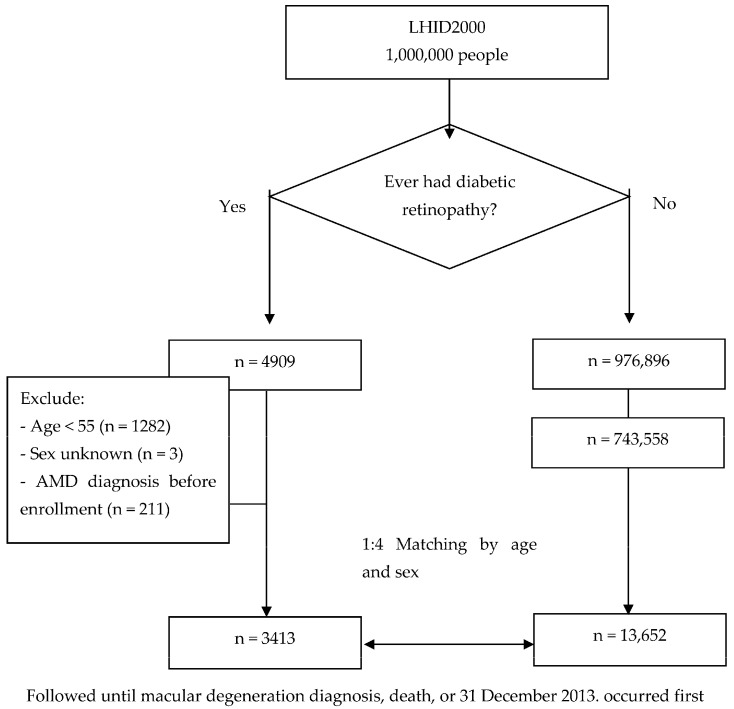
Study flowchart. AMD = age-related macular degeneration, LHID = Longitudinal Health Insurance Database.

**Figure 2 biomedicines-12-00727-f002:**
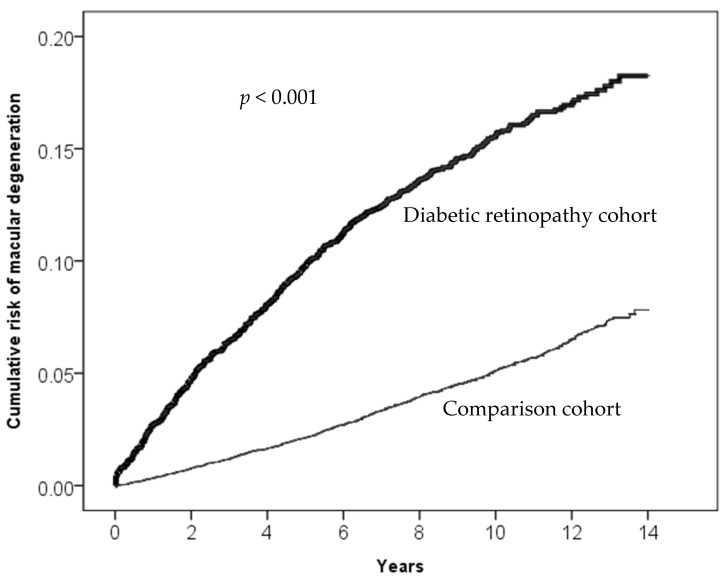
Cumulative risk of all types of AMD (age-related macular degeneration) (ICD-9 code 362.50, 362.51, and 362.52) in the DR group and the non-DR group.

**Figure 3 biomedicines-12-00727-f003:**
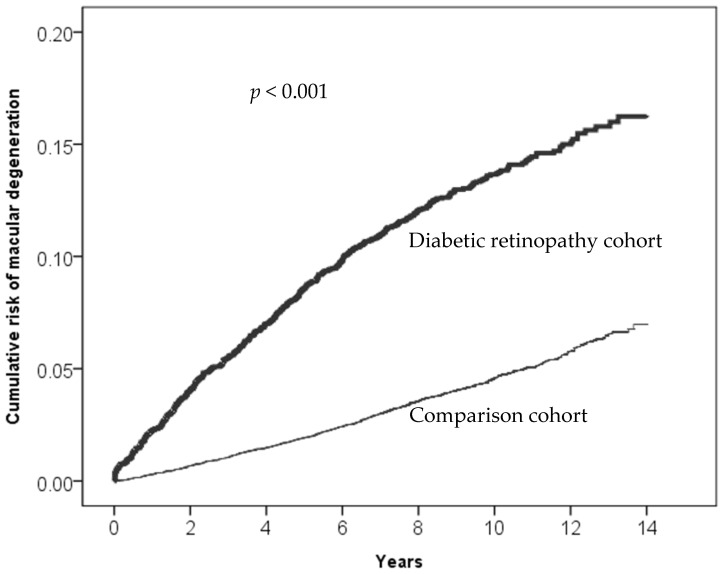
Cumulative risk of senile (unspecified) age-related macular degeneration (ICD-9 code 362.50) in the diabetic retinopathy cohort and the comparison cohort.

**Figure 4 biomedicines-12-00727-f004:**
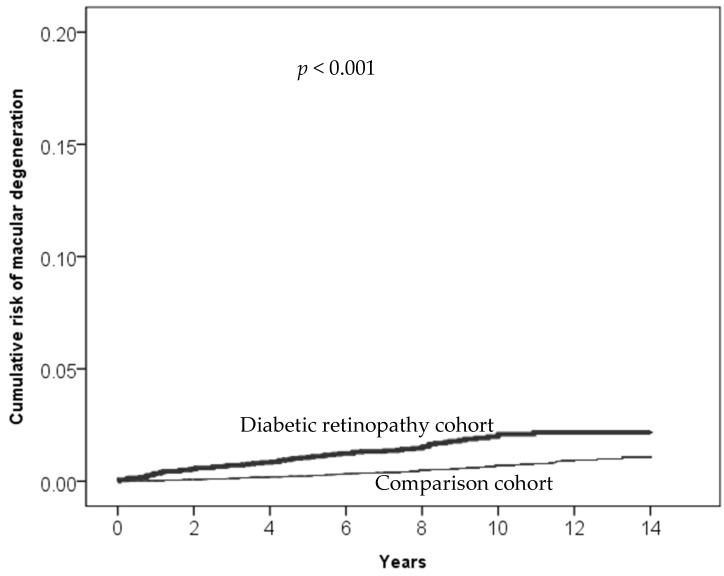
Cumulative risk of non-exudative age-related macular degeneration (ICD-9 code 362.51) in the diabetic retinopathy cohort and the comparison cohort.

**Figure 5 biomedicines-12-00727-f005:**
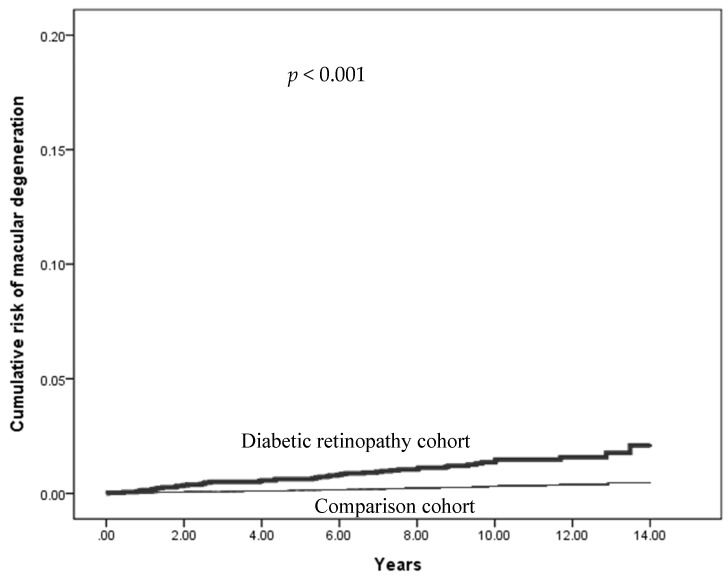
Cumulative risk of exudative age-related macular degeneration (ICD-9 code 362.52) in the diabetic retinopathy cohort and the comparison cohort.

**Table 1 biomedicines-12-00727-t001:** Baseline demographic status and comorbidity comparison between non-diabetic retinopathy and diabetic retinopathy groups.

Variable	Non-DR Group *N* = 13,652 (%)	DR Group *N* = 3413 (%)	*p*-Value
Age, years (SD) *	66.5 (7.9)	67.1 (7.2)	<0.001
55–65	5812 (42.6)	1453 (42.6)	1.000
65–75	5832 (42.7)	1458 (42.7)	
75–85	1868 (13.7)	467 (13.7)	
≥85	140 (1.0)	35 (1.0)	
Sex			1.000
Female	7732 (56.6)	1933 (56.6)	
Male	5920 (43.4)	1480 (43.4)	
Comorbidity			
Hypertension	6252 (45.8)	2354 (69.0)	<0.001
Hyperlipidemia	2593 (19.0)	950 (27.8)	<0.001
CAD	2412 (17.7)	1356 (39.7)	<0.001
Stroke	1405 (10.3)	1029 (30.2)	<0.001
COPD	3955 (29.0)	1265 (37.1)	<0.001
Liver cirrhosis and chronic hepatitis	1566 (11.5)	601 (17.6)	<0.001

* Independent *t*-test. Abbreviations: SD: standard deviation; non-DR: non-diabetic retinopathy; DR: diabetic retinopathy; CAD: coronary artery disease; COPD: chronic obstructive pulmonary disease.

**Table 2 biomedicines-12-00727-t002:** Incidence and estimated hazard ratios of all types of age-related macular degeneration (AMD) (ICD-9 codes 362.50, 362.51, 362.52).

Variable	Non-DR Group			DR Group			HR (95% CI)	*p*-Value	aHR (95% CI)	*p*-Value
	Event	PYs	Rate	Event	PYs	Rate				
DR	704	131,328	5.36	514	30,644	16.77	3.14 (2.80–3.52)	<0.001	3.50 (3.10–3.95)	<0.001
Demographic										
Age group										
55–65	252	57,141	4.41	218	13,412	16.25	3.70 (3.09–4.44)	<0.001	4.17 (3.44–5.06)	<0.001
65–75	380	56,031	6.78	210	13,117	16.01	2.37 (2.00–2.81)	<0.001	2.62 (2.19–3.12)	<0.001
75–85	70	16,959	4.13	81	3805	21.29	5.06 (3.67–6.96)	<0.001	5.69 (4.02–8.07)	<0.001
≥85	2	1196	1.67	5	311	16.08	10.18 (1.97–52.53)	0.006	16.64 (1.29–214.78)	0.031
Sex										
Female	451	74,920	6.02	297	17,648	16.83	2.81 (2.42–3.25)	<0.001	3.14 (2.69–3.66)	<0.001
Male	253	56,408	4.49	217	12,996	16.70	3.72 (3.10–4.46)	<0.001	4.15 (3.42–5.04)	<0.001
Comorbidity										
Hypertension										
No	311	71,194	4.37	210	9739	21.56	4.95 (4.15–5.89)	<0.001	5.44 (4.53–6.53)	<0.001
Yes	393	60,133	6.54	304	20,905	14.54	2.24 (1.93–2.60)	<0.001	2.57 (2.20–3.00)	<0.001
Hyperlipidemia										
No	569	106,311	5.35	408	22,157	18.41	3.44 (3.03–3.91)	<0.001	3.71 (3.25–4.25)	<0.001
Yes	135	25,016	5.40	106	8488	12.49	2.34 (1.82–3.02)	<0.001	2.68 (2.06–3.49)	<0.001
CAD										
No	534	108,276	4.93	356	18,627	19.11	3.88 (3.39–4.43)	<0.001	4.24 (3.69–4.88)	<0.001
Yes	170	23,052	7.37	158	12,017	13.15	1.80 (1.45–2.23)	<0.001	2.11 (1.69–2.63)	<0.001
Stroke										
No	657	118,136	5.56	457	21,689	21.07	3.80 (3.37–4.28)	<0.001	3.72 (3.29–4.22)	<0.001
Yes	47	13,191	3.56	57	8955	6.37	1.80 (1.22–2.65)	0.003	1.75 (1.17–2.60)	0.006
COPD										
No	460	93,850	4.90	356	19,227	18.52	3.77 (3.29–4.34)	<0.001	4.19 (3.62–4.87)	<0.001
Yes	244	37,478	6.51	158	11,417	13.84	2.14 (1.75–2.62)	<0.001	2.45 (1.99–3.01)	<0.001
Liver cirrhosis and chronic hepatitis										
No	616	116,796	5.27	429	25,350	16.92	3.21 (2.84–3.64)	<0.001	3.56 (3.12–4.05)	<0.001
Yes	88	14,531	6.06	85	5294	16.06	2.69 (2.00–3.62)	<0.001	3.11 (2.29–4.24)	<0.001

Model adjusted for age, sex, low income, hypertension, hyperlipidemia, CAD, stroke, COPD, and liver cirrhosis and chronic hepatitis. Abbreviations: DR: diabetic retinopathy; AMD: age-related macular degeneration; CAD: coronary artery disease; COPD: chronic obstructive pulmonary disease; PYs: person years; rate: incidence rate per 10,000 person years; HR: hazard ratio; aHR: adjusted hazard ratio.

**Table 3 biomedicines-12-00727-t003:** Incidence and estimated hazard ratios of senile (unspecified) age-related macular degeneration (AMD) (ICD-9 code 362.50).

Variable	Non-DR Group			DR Group			HR (95% CI)	*p*-Value	aHR (95% CI)	*p*-Value
	Event	PYs	Rate	Event	PYs	Rate				
DR	629	131,665	4.78	455	31,053	14.65	3.08 (2.73–3.47)	<0.001	3.45 (3.04–3.92)	<0.001
Demographic										
Age group										
55–65	225	57,239	3.93	191	13,591	14.05	3.58 (2.96–4.35)	<0.001	4.04 (3.29–4.97)	<0.001
65–75	342	56,210	6.08	185	13,309	13.90	2.30 (1.92–2.75)	<0.001	2.55 (2.12–3.08)	<0.001
75–85	60	17,020	3.53	76	3830	19.84	5.51 (3.93–7.74)	<0.001	6.19 (4.28–8.94)	<0.001
≥85	2	1196	1.67	3	322	9.32	5.95 (0.99–35.64)	0.051	4.16 (0.28–61.35)	0.298
Sex										
Female	411	75,133	5.47	269	17,840	15.08	2.77 (2.37–3.23)	<0.001	3.09 (2.63–3.34)	<0.001
Male	218	56,532	3.86	186	13,213	14.08	3.65 (3.00–4.44)	<0.001	4.12 (3.35–5.07)	<0.001
Comorbidity										
Hypertension										
No	271	71,318	3.80	188	9833	19.12	5.05 (4.19–6.08)	<0.001	5.57 (4.67–6.89)	<0.001
Yes	358	60,348	5.93	267	21,220	12.58	2.13 (1.82–2.50)	<0.001	2.45 (2.08–2.89)	<0.001
Hyperlipidemia										
No	499	106,574	4.68	362	22,457	16.12	3.45 (3.02–3.95)	<0.001	3.76 (3.26–4.34)	<0.001
Yes	130	25,091	5.18	93	8596	10.82	2.11 (1.62–2.75)	<0.001	2.41 (1.83–3.18)	<0.001
CAD										
No	481	108,488	4.43	314	18,803	16.70	3.77 (3.27–4.35)	<0.001	4.14 (3.57–4.80)	<0.001
Yes	148	23,177	6.39	141	12,249	11.51	1.82 (1.44–2.29)	<0.001	2.13 (1.68–2.69)	<0.001
Stroke										
No	587	118,404	4.96	408	21,952	18.59	3.76 (3.31–4.27)	<0.001	3.69 (3.23–4.21)	<0.001
Yes	42	13,262	3.17	47	9101	5.16	1.64 (1.08–2.49)	0.020	1.56 (1.01–2.40)	0.043
COPD										
No	416	94,032	4.42	316	19,484	16.22	3.67 (3.17–4.25)	<0.001	4.06 (3.47–4.75)	<0.001
Yes	213	37,633	5.66	139	11,569	12.01	2.14 (1.73–2.65)	<0.001	2.47 (1.98–3.07)	<0.001
Liver cirrhosis and chronic hepatitis										
No	555	117,040	4.74	376	25,674	14.65	3.10 (2.72–3.53)	<0.001	3.47 (3.02–3.98)	<0.001
Yes	74	14,625	5.06	79	5379	14.69	2.94 (2.14–4.04)	<0.001	3.30 (2.38–4.59)	<0.001

Model adjusted for age, sex, low income, hypertension, hyperlipidemia, CAD, stroke, COPD, and liver cirrhosis and chronic hepatitis. Abbreviations: DR: diabetic retinopathy; AMD: age-related macular degeneration; CAD: coronary artery disease; COPD: chronic obstructive pulmonary disease; PYs: person years; rate: incidence rate per 10,000 person years; HR: hazard ratio; aHR: adjusted hazard ratio.

**Table 4 biomedicines-12-00727-t004:** Incidence and estimated hazard ratios of non-exudative age-related macular degeneration (AMD) (ICD-9 code 362.51).

Variable	Non-DR Group			DR Group			HR (95% CI)	*p*-Value	aHR (95% CI)	*p*-Value
	Event	PYs	Rate	Event	PYs	Rate				
DR	93	134,162	0.69	63	33,404	1.89	2.73 (1.98–3.76)	<0.001	2.92 (2.08–4.09)	<0.001
Demographic										
Age group										
55–65	29	58,124	0.50	26	14,503	1.79	3.59 (2.12–6.10)	<0.001	4.17 (2.37–7.34)	<0.001
65–75	51	57,578	0.89	26	14,321	1.82	2.06 (1.28–3.30)	0.003	2.06 (1.27–3.37)	0.004
75–85	13	17,247	0.75	9	4254	2.12	2.79 (1.19–6.53)	0.018	3.65 (1.45–9.18)	0.006
≥85	0	1213	0.00	2	326	6.13	NA	NA	NA	NA
Sex										
Female	52	76,773	0.68	30	19,251	1.56	2.31 (1.47–3.62)	0.001	2.51 (1.56–4.02)	0.001
Male	41	57,389	0.71	33	14,154	2.33	3.27 (2.07–5.18)	<0.001	3.45 (2.11–5.62)	<0.001
Comorbidity										
Hypertension										
No	43	71,997	0.60	25	10,349	2.42	4.07 (2.49–6.67)	<0.001	3.97 (2.36–6.67)	<0.001
Yes	50	62,164	0.80	38	23,055	1.65	2.05 (1.35–3.14)	0.001	2.40 (1.55–3.70)	<0.001
Hyperlipidemia										
No	79	108,192	0.73	47	24,076	1.95	2.67 (1.86–3.84)	<0.001	2.84 (1.94–4.16)	<0.001
Yes	14	25,970	0.54	16	9328	1.72	3.22 (1.57–6.59)	0.001	3.18 (1.51–6.69)	0.002
CAD										
No	65	110,205	0.59	41	20,028	2.05	3.48 (2.35–5.14)	<0.001	3.84 (2.59–5.77)	<0.001
Yes	28	23,957	1.17	22	13,376	1.64	1.42 (0.81–2.48)	0.217	1.71 (0.96–3.02)	0.067
Stroke										
No	88	120,519	0.73	57	23,436	2.43	3.35 (2.40–4.67)	<0.001	3.07 (2.17–4.36)	<0.001
Yes	5	13,642	0.37	6	9969	0.60	1.65 (0.50–5.41)	0.409	1.44 (0.43–4.87)	0.557
COPD										
No	55	95,300	0.58	38	20,842	1.82	3.17 (2.10–4.79)	<0.001	3.63 (2.33–5.67)	<0.001
Yes	38	38,861	0.98	25	12,562	1.99	2.04 (1.23–3.39)	0.006	2.23 (1.32–3.75)	0.003
Liver cirrhosis and chronic hepatitis										
No	75	119,043	0.63	53	27,608	1.92	3.06 (2.15–4.35)	<0.001	3.18 (2.19–4.62)	<0.001
Yes	18	15,118	1.19	10	5796	1.73	1.46 (0.67–3.15)	0.341	1.86 (0.84–4.15)	0.127

Model adjusted for age, sex, low income, hypertension, hyperlipidemia, CAD, stroke, COPD, and liver cirrhosis and chronic hepatitis. Abbreviations: DR: diabetic retinopathy; AMD: age-related macular degeneration; CAD: coronary artery disease; COPD: chronic obstructive pulmonary disease; PYs: person years; rate: incidence rate per 10,000 person years; HR: hazard ratio; aHR: adjusted hazard ratio; NA: not applicable.

**Table 5 biomedicines-12-00727-t005:** Incidence and estimated hazard ratios of exudative age-related macular degeneration (AMD) (ICD-9 code 362.52).

Variable	Non-DR Group			DR Group			HR (95% CI)	*p*-Value	aHR (95% CI)	*p*-Value
	Event	PYs	Rate	Event	PYs	Rate				
DR	42	134,395	0.31	46	33,545	1.37	4.38 (2.88–6.66)	<0.001	3.92 (2.51–6.14)	<0.001
Demographic										
Age group										
55–65	13	58,166	0.22	15	14,571	1.03	4.65 (2.21–9.78)	<0.001	4.20 (1.88–9.38)	0.001
65–75	22	57,716	0.38	24	14,371	1.67	4.33 (2.43–7.73)	<0.001	3.63 (1.97–6.71)	<0.001
75–85	7	17,300	0.40	7	4265	1.64	4.01 (1.41–11.45)	0.009	3.98 (1.32–12.02)	0.014
≥85	0	1213	0.00	0	338	0.00	NA	NA	NA	NA
Sex										
Female	21	76,917	0.27	20	19,315	1.04	3.83 (2.08–7.07)	<0.001	3.16 (1.65–6.07)	0.001
Male	21	57,478	0.37	26	14,230	1.83	4.98 (2.80–8.86)	<0.001	4.72 (2.55–8.74)	<0.001
Comorbidity										
Hypertension										
No	17	72,052	0.24	12	10,407	1.15	4.87 (2.33–10.21)	<0.001	4.67 (2.14–10.16)	0.001
Yes	25	62,343	0.40	34	23,138	1.47	3.66 (2.18–6.14)	<0.001	3.57 (2.09–6.10)	<0.001
Hyperlipidemia										
No	36	108,329	0.33	34	24,200	1.40	4.19 (2.62–6.70)	<0.001	3.77 (2.29–6.23)	<0.001
Yes	6	26,066	0.23	12	9345	1.28	5.68 (2.13–15.15)	0.001	4.66 (1.68–12.95)	0.003
CAD										
No	28	110,368	0.25	21	20,137	1.04	4.14 (2.35–7.28)	<0.001	4.11 (2.28–7.42)	<0.001
Yes	14	24,027	0.58	25	13,408	1.86	3.17 (1.65–6.11)	0.001	3.46 (1.78–6.75)	0.001
Stroke										
No	37	120,695	0.31	36	23,607	1.52	4.95 (3.13–7.84)	<0.001	4.19 (2.59–6.79)	<0.001
Yes	5	13,700	0.36	10	9938	1.01	2.78 (0.95–8.12)	0.062	2.71 (0.90–8.18)	0.077
COPD										
No	25	95,498	0.26	33	20,902	1.58	6.04 (3.59–10.15)	<0.001	4.64 (2.64–8.14)	<0.001
Yes	17	38,897	0.44	13	12,643	1.03	2.33 (1.13–4.80)	0.022	2.67 (1.27–5.64)	0.011
Liver cirrhosis and chronic hepatitis										
No	40	119,194	0.34	38	27,690	1.37	4.07 (2.61–6.34)	<0.001	3.41 (2.12–5.48)	<0.001
Yes	2	15,200	0.13	8	5856	1.37	10.53 (2.24–49.60)	0.003	12.02 (2.47–58.45)	0.002

Model adjusted for age, sex, low income, hypertension, hyperlipidemia, CAD, stroke, COPD, and liver cirrhosis and chronic hepatitis. Abbreviations: DR: diabetic retinopathy; AMD: age-related macular degeneration; CAD: coronary artery disease; COPD: chronic obstructive pulmonary disease; PYs: person years; rate: incidence rate per 10,000 person years; HR: hazard ratio; aHR: adjusted hazard ratio; NA: not applicable.

**Table 6 biomedicines-12-00727-t006:** ICD-9-CM Volume 2 Index entries containing backreference to 362.50; degeneration and degenerative.

Macula (Senile) (Unspecified) 362.50.
- non-exudative 362.51
- exudative 362.52
- atrophic 362.51
- best’s 362.76
- congenital 362.75
- cystic 362.54
- cystoid 362.53
- disciform 362.52
- dry 362.51
- familial pseudoinflammatory 362.77
- hereditary 362.76
- hole 362.54
- juvenile 362.75
- pseudohole 362.54
- wet 362.52

## Data Availability

Not applicable.

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
