# Peer review of "The Association between Diabetic Retinopathy and Macular Degeneration: A Nationwide Population-Based Study"

_biomedicines, 2024, doi:10.3390/biomedicines12040727_

Round 1

Reviewer 1 Report

Comments and Suggestions for Authors

Manuscript # biomedicines-2879712

Title: The Association between diabetic retinopathy and macular degeneration: a nationwide population-based study

Major revision

Authors used the Taiwan National Health Insurance database and performed a nationwide population-based study to examine the association between DR and AMD. Authors conclude that DR is a significant risk factor for exudative and non-exudative AMD after adjusting for demographic and comorbid conditions.

This is an important study and acceptable for publication in Biomedicines. However, one concern is existed in the manuscript. The classification of types of AMD in this study is not clear. What is a senile AMD? The results of all types of AMD and exudative or non-exudative AMD are largely different (Figs 2-5). Authors must clarify the definition of AMD in details in this study and prepare the table to display numbers of all types of AMD respectively. Otherwise, readers cannot understand the significance of this study precisely. This revision is essential.

In Figure 5, the position of “P<0.001” is out of alignment. Please adjust it.

End of comments

Author Response

Reviewer 1.

Major revision 

Authors used the Taiwan National Health Insurance database and performed a nationwide population-based study to examine the association between DR and AMD. Authors conclude that DR is a significant risk factor for exudative and non-exudative AMD after adjusting for demographic and comorbid conditions. This is an important study and acceptable for publication in Biomedicines. However, one concern is existed in the manuscript.

  1. The classification of types of AMD in this study is not clear. What is a senile AMD? The results of all types of AMD and exudative or non-exudative AMD are largely different (Figs 2-5).

Answer: We appreciate your excellent recommendation and suggestion. Following the ICD-9-CM format, there are several types of macular degeneration (Table 6) including senile(unspecified), non-exudative, exudative, familial, juvenile, congenital, cystic AMD etc. and we chose the most often cited codes by Ophthalmologist, that is, senile (unspecified, 362.50). non-exudative (362.51) and exudative (362.52) AMD to represent all types of AMD for convenience of further analysis. Most Ophthalmologists cited senile (unspecified) AMD which should include non-exudative and exudative AMD that result in large difference in patient number among different types of AMD. This inconsistency of citation by Ophthalmologists might result in large number difference among different types of AMD. (Please see Study Population, P6, L2-8). We also add this discrepancy during coding to discussion section as study limitation. (Discussion, P12, L1-5)

  1. Authors must clarify the definition of AMD in details in this study and prepare the table to display numbers of all types of AMD respectively. Otherwise, readers cannot understand the significance of this study precisely. This revision is essential.

Answer: Thank you very much for your great recommendation. This is a retrospective study analyzing data from Longitudinal National Insurance Database (LNID). Following the ICD-9-CM format, there are several types of macular degeneration (table 6) including senile(unspecified), non-exudative, exudative, familial, juvenile, congenital, cystic AMD etc. We chose the most often cited codes by Ophthalmologist, that is, senile (unspecified, 362.50). non-exudative (362.51) and exudative (362.52) AMD to represent all types of AMD for convenience of further analysis. From Table 2 to 5, we had already listed AMD numbers as events item from all type, senile(unspecified), non-exudative, exudative AMD type. In addition, we use patient year instead of patient number for more accurate analysis because each patient has different duration of follow up.

In Figure 5, the position of “P<0.001” is out of alignment. Please adjust it.

Answer: We adjust as your suggestion.

Reviewer 2 Report

Comments and Suggestions for Authors

This paper assesses the risk of developing overall, non-exudative, and exudative AMD in individuals with diabetic retinopathy (DR) compared to those without retinopathy (non-DR) based on a Taiwan nationwide population study. The topic is valuable and the sample size is large, but the use of English in the article is not appropriate, the logic is not very smooth, the research design is not very clear and convinsive. I hope that it can be carefully revised, with specific issues as follows:

1. In the abstract, the objective is unreasonable, 'As diabetes becomes increasingly available, it is crucial to explore the potential relation between diabetes-related complications, such as diabetic retinopathy (DR), and AMD.' Firstly, 'available' is not appropriate here. Furthermore, AMD is not a diabetes-related complication. The conclusion, 'DR patients tend to have a higher prevalence of vascular comorbidities; however, our findings indicate that the ocular pathologies inherent to DR might have a more significant impact on the progression to AMD,' cannot be drawn solely from the results of this article.

2. Improper use of citations.

3. In the Introduction section, 'A relation between DM and early stages of AMD was noted among Korean elderly patients; whereas, a multicenter, population-based European study found a positive relation between DM and advanced stages of AMD.' Please specify the exact trend.

4. The explanation of the study population groupings is unclear. Does it exclude patients who were previously diagnosed with DR or DM, or does it include patients with DR or DM? Why not analyze patients with diabetes without DR as a separate group? Why assign four non-DR patients for comparison with one DR patient, and what is the purpose? Is one patient included with one eye or two eyes? If two eyes, nesting eyes within patients should be considered for regression analyses.

5.Why are 'A history of comorbidities' used as an adjustment factor, rather than other factors more closely related to the incidence of DR and AMD, such as the type of DM, the duration of DM, smoking, etc.?

6.Please present the p-values of all statistical results.

7.Why use 'senile AMD' when simply 'AMD' would suffice?

8.Given that past research has confirmed that the epidemiological relationship between DR and AMD is still a subject of debate, the conclusion should discuss the similarities and differences between the results of this study and those of other studies, the related possibilities, and the directions for future research.

Comments on the Quality of English Language

Sometimes the language and word choice are not quite appropriate, and the logic is not very sound, which makes me wonder if it's an issue of not being sufficiently proficient in English expression.

Author Response

This paper assesses the risk of developing overall, non-exudative, and exudative AMD in individuals with diabetic retinopathy (DR) compared to those without retinopathy (non-DR) based on a Taiwan nationwide population study. The topic is valuable and the sample size is large, but the use of English in the article is not appropriate, the logic is not very smooth, and the research design is not very clear and convinsive. I hope that it can be carefully revised, with specific issues as follows:

  1. In the abstract, the objective is unreasonable, 'As diabetes becomes increasingly available, it is crucial to explore the potential relation between diabetes-related complications, such as diabetic retinopathy (DR), and AMD.' Firstly, 'available' is not appropriate here. Furthermore, AMD is not a diabetes-related complication. The conclusion, 'DR patients tend to have a higher prevalence of vascular comorbidities; however, our findings indicate that the ocular pathologies inherent to DR might have a more significant impact on the progression to AMD,' cannot be drawn solely from the results of this article.

Answer: Thank you very much for your excellent recommendation and suggestion which greatly promote the quality of our manuscript. We had requested for English language correction by MDPI according to editor’s suggestion. We change “As diabetes becomes increasingly available…” into “As diabetes becomes increasingly prevalent in aging…” (Abstract P2, L4). We agree with your comment that AMD is not complication of diabetes and delete the phrase “diabetes-related complications, such as” (Abstract P2, L5). We also re-write the sentence in introduction (Introduction P4, L12-14) and delete the diabetic complications as follows: “These inflammatory and vascular changes are common mechanisms underlying pivotal in the progression of diabetic complications, specifically DR and AMD [4-6].”

  1. Improper use of citations.

Answer: Thank you for pointing this out. We have meticulously reviewed and corrected our citations to ensure accuracy and relevance. In order to avoid confusion to readers, we delete following 4-references:

  1. Leske, M.C.; Wu, S.Y.; Hennis, A.; Nemesure, B.; Yang, L.; Hyman, L.; Schachat, A.P.; Barbados Eye Studies, G. Nine-year incidence of age-related macular degeneration in the Barbados Eye Studies. Ophthalmology 2006, 113, 29-35, doi:10.1016/j.ophtha.2005.08.012.
  2. Klein, R.; Klein, B.E.; Moss, S.E. Diabetes, hyperglycemia, and age-related maculopathy. The Beaver Dam Eye Study. Ophthalmology 1992, 99, 1527-1534.
  3. Rofagha, S.; Bhisitkul, R.B.; Boyer, D.S.; Sadda, S.R.; Zhang, K.; Group, S.-U.S. Seven-year outcomes in ranibizumab-treated patients in ANCHOR, MARINA, and HORIZON: a multicenter cohort study (SEVEN-UP). Ophthalmology 2013, 120, 2292-2299, doi:10.1016/j.ophtha.2013.03.046.
  4. Ardeljan, D.; Chan, C.C. Aging is not a disease: distinguishing age-related macular degeneration from aging. Prog Retin Eye Res 2013, 37, 68-89, doi:10.1016/j.preteyeres.2013.07.003.

  1. In the Introduction section, 'A relation between DM and early stages of AMD was noted among Korean elderly patients; whereas, a multicenter, population-based European study found a positive relation between DM and advanced stages of AMD.' Please specify the exact trend.

Answer: Thank you for your recommendation. To specify the similar and exact trend in both studies, we re-write the sentence as following: DM itself related positively with early AMD among elderly Korean patients [20], whereas similar relation was noted between DR and neovascular AMD in a multicenter, population-based European study [21].

  1. The explanation of the study population groupings is unclear. Does it exclude patients who were previously diagnosed with DR or DM, or does it include patients with DR or DM? Why not analyze patients with diabetes without DR as a separate group? Why assign four non-DR patients for comparison with one DR patient, and what is the purpose? Is one patient included with one eye or two eyes? If two eyes, nesting eyes within patients should be considered for regression analyses.

Answer: Thank you for your great recommendation and suggestion. We exclude patients who were previously diagnosed with DM, DR or AMD before the index date of 2000. You have raised a great comment to have one group with DM while without DR. This idea might be conducted in our future work. We selected four age and sex matched non-DR patients for comparison for one DR patient to strengthen statistic significancy. From the ICD-9 code record, we did not know whether one or two eyes were diagnosed. Most patients cited one ICD-9 code while they can cite up to three ICD-9 codes even though two eyes had been diagnosed. In this study, Kaplan–Meier curves were used to depict cumulative AMD incidence and Cox regression was used to estimate AMD risk, adjusted for age, sex, and comorbidities and stratified by age and sex to determine association between DR and AMD.

5.Why are 'A history of comorbidities' used as an adjustment factor, rather than other factors more closely related to the incidence of DR and AMD, such as the type of DM, the duration of DM, smoking, etc.?

Answer: As we mentioned in discussion section (Discussion, P10, L5-10), “comorbidities such as hypertension, hyperlipidemia, and coronary artery disease (CAD) was observed in patients with DR. These findings are in line with previous research underscoring the multifaceted nature of diabetes, with its extensive impact on various systemic diseases. These vascular comorbidities are known to contribute to damage in retinal small vessel endothelial cells, thereby influencing the development and severity of DR and might also have impact on the development of AMD.” Adjusting for these comorbidities in the analysis not only enhances the robustness of the findings but also strengthens the evidence of the positive association between DR and AMD risk (Discussion, strength of our study, P 11, L22-24). We therefore used a history of comorbidities as an adjustment factor. Smoking was not included in Longitudinal National Insurance Database (LNID). We selected patients (aged ≥55 years old) who were newly diagnosed with diabetic retinopathy (DR) using ICD-9-CM codes which did not include most type 1 DM. Duration of DM was not included in ICD-9 database.

6.Please present the p-values of all statistical results.

Answer: We had presented the p-values of all statistical results.

7.Why use 'senile AMD' when simply 'AMD' would suffice?

Answer: Thank you very much for your great recommendation. Following the ICD-9-CM format, there are several types of macular degeneration (table 6) including senile(unspecified), non-exudative, exudative, familial, juvenile, congenital, cystic AMD etc. and we chose the most often cited codes by Ophthalmologist, that is, senile (unspecified, 362.50) AMD. non-exudative (362.51) AMD and exudative (362.52) AMD to represent all types of AMD for convenience of further analysis. So, we used senile (unspecified) AMD according to the ICD-9 classifications to distinguish with AMD.

8.Given that past research has confirmed that the epidemiological relationship between DR and AMD is still a subject of debate, the conclusion should discuss the similarities and differences between the results of this study and those of other studies, the related possibilities, and the directions for future research. 

Answer: Thank you for your recommendation on conclusion. We modify and re-write our conclusion as follows: (Discussion, P12, L8-22)

“In conclusion, our study revealed a multifaceted relationship between DR and AMD. DR patients are more likely to have vascular conditions which can contribute to the onset of all types of AMD. Here, we firstly provided the relation between DR and AMD regardless of co-morbid conditions, as to suggest that the inherent pathologies related to DR might have a greater impact on the progression to AMD than comorbid conditions. This is in contrast to the non-DR cohort, where a clearer correlation between comorbidities and the risk of AMD is observed. However, our study diverges in its emphasis on the heightened risk of non-exudative AMD in DR patients, a less explored aspect in previous studies. This discrepancy underscores the complexity of their association and the imperative for continued exploration in this field. Nevertheless, the study highlights the critical need for regular ocular monitoring in individuals with DR, with a particular focus on detecting AMD. Healthcare providers should maintain a heightened awareness of the increased risk of AMD in the DR population and emphasize the need for proactive and preventive eye care measures.”

Round 2

Reviewer 1 Report

Comments and Suggestions for Authors

Authors answered all questions and revised the manuscript completely. This version is acceptable for publication in Biomedicines.

Reviewer 2 Report

Comments and Suggestions for Authors

The quality of the manuscript has been greatly improved.